# Exploring the Role of Physical Activity in Mediating the Association between Educational Level and Health-Related Quality of Life in an Adult Lifespan Sample from Madeira Island

**DOI:** 10.3390/ijerph19137608

**Published:** 2022-06-22

**Authors:** Jesús García-Mayor, Élvio Rúbio Gouveia, Adilson Marques, Ernesto De la Cruz-Sánchez, Antonio Moreno-Llamas, Cíntia França, Bruna R. Gouveia, Andreas Ihle

**Affiliations:** 1Public Health and Epidemiology Research Group, San Javier Campus, University of Murcia, 30720 San Javier, Spain; jesus.garcia9@um.es (J.G.-M.); erneslacruz@um.es (E.D.l.C.-S.); antonio.moreno13@um.es (A.M.-L.); 2Department of Physical Education and Sport, University of Madeira, 9020-105 Funchal, Portugal; cintia.franca@staff.uma.pt; 3LARSYS, Interactive Technologies Institute, 9020-105 Funchal, Portugal; bruna.gouveia@madeira.gov.pt; 4Center for the Interdisciplinary Study of Gerontology and Vulnerability, University of Geneva, 1205 Geneva, Switzerland; andreas.ihle@unige.ch; 5Interdisciplinary Centre for the Study of Human Performance (CIPER), Faculty of Human Kinetics, University of Lisbon, 1499-002 Lisbon, Portugal; amarques@fmh.ulisboa.pt; 6Instituto de Saúde Ambiental (ISAMB), University of Lisbon, 1649-020 Lisbon, Portugal; 7Regional Directorate of Health, Secretary of Health of the Autonomous Region of Madeira, 9004-515 Funchal, Portugal; 8Saint Joseph of Cluny Higher School of Nursing, 9050-535 Funchal, Portugal; 9Department of Psychology, University of Geneva, 1205 Geneva, Switzerland; 10Swiss National Centre of Competence in Research LIVES—Overcoming Vulnerability: Life Course Perspectives, 1015 Lausanne, Switzerland

**Keywords:** exercise, functioning, mediation analysis, perceived health, socioeconomic status, sport

## Abstract

**Background:** People more socioeconomically vulnerable tend to have a poorer health-related quality of life (HRQoL). Studies are trying to analyse the factors that may condition this relationship, including physical activity (PA), which may influence the relationship between socioeconomic status (SES) and HRQoL. This study aimed to analyse the relationship between SES and HRQoL through specific domains of PA. **Methods:** A total of 381 adults (≥18 years) from the Autonomous Region of Madeira completed the measurements. Mediation analyses using bootstrapping methods adjusted for confounding variables were performed to relate SES and HRQoL; the latter was evaluated using the physical component score (PCS) of the SF-12, the mental component score (MCS) of the SF-12, and the total score in the SF-12 questionnaire (SF-12 score), through physical activity. **Results:** Educational level was positively related to PCS and SF-12 score. Leisure-time PA (not including sports PA) and PA at work, as single mediating variables, did not mediate the relationship between SES and HRQoL. The total PA suppressed the socioeconomic gradient of HRQoL by 8–10%, 39–46%, and 15–16%, respectively, for the PCS, MCS, and the SF-12 score; sports PA mediated the relationship by 13–16%, 50%, and 15–21%, respectively. **Conclusion****s****:** The results suggest that sports PA contributes to reducing the socioeconomic gradient of HRQoL.

## 1. Introduction

Health-related quality of life (HRQoL) is a concept that refers to the assessment of health perception and is often conceived of as health in broader terms, using functioning and well-being rather than clinical measures [1]. Due to significant disabilities and increasing disability-adjusted life years [2], driven mainly by the rise of non-communicable diseases [3], preventing the worsening HRQoL of the population has become a public health challenge. Therefore, there is a constant and persistent search to identify the determinants of HRQoL.

Socioeconomic status (SES), referring to the position a person occupies in the structure of society due to economic and social factors [4], is a determinant of HRQoL [5]. Frequently, people with lower SES have a higher risk of the incidence of non-communicable diseases, including cardiovascular diseases, cancer, respiratory diseases, or diabetes [6]. The higher incidence of these non-communicable diseases could be influenced by the higher prevalence of unhealthy lifestyle behaviours, more common in people with lower SES, such as smoking, alcohol use, unhealthy eating, and lower physical activity (PA) [7].

Regular PA has been shown to contribute to a lower risk of non-communicable diseases and mortality and improved physical, social, and mental well-being [8]. Regarding substantial health benefits for the population related to PA, the WHO has established PA guidelines for the adult population, recommending at least 150–300 min of moderate-intensity aerobic PA, or at least 75–150 min of vigorous-intensity aerobic PA, or an equivalent combination of moderate- and vigorous-intensity activity throughout the week [8]. There is strong evidence that people who comply with these recommendations show better health and well-being measures [9,10,11,12], and the benefits of PA level on HRQoL have been reported [9]. However, a “PA paradox” shows that not all types of PA positively affect health and well-being [13,14,15]. Thus, according to the specific domains of PA, total [16,17] and leisure-time PA [18,19] are positively related to HRQoL. In contrast, job-related/work PA [20] has been shown to have a negative effect. The negative results of specific domains highlight the importance of the type and setting of PA [15].

Meanwhile, the limitation of not using different domains of PA to estimate the association with HRQoL has been previously discussed [21]. These specific PA domains also follow a different socioeconomic gradient. For example, it reports that those with higher SES are more physically active during leisure time (including sports PA) than those with lower SES [22,23]. Additionally, job-related/work PA is more prevalent in people with lower SES [23]. Despite this strong evidence regarding leisure and job-related/work PA, the social gradient of total PA does not have a pattern as consistent as the previous ones. [23]. Therefore, studies investigating the relationship between SES and HRQoL should incorporate the possible interaction of PA. Furthermore, considering that not all types of PA have a similar relationship with SES and HRQoL, this possible interaction should consider PA’s domain-specific (work, sport, leisure time, and total) characteristics. To target these gaps, in the present study we examine the relationship between SES and HRQoL, taking into account the mediating effect of PA in adults over 18 years of age residing in the Autonomous Region of Madeira (Portugal).

## 2. Materials and Methods

### 2.1. Sample and Study Design

This cross-sectional study included 381 participants (aged 18 to 89 years old). The sample comprises an active population affiliated with the Madeira Association of Sport for All in the Autonomous Region of Madeira, Funchal, Portugal [24]. Participants were volunteers recruited to participate between January and August 2017 through direct contacts in gyms, cultural and sports clubs, and associations that offer sports for all activities. Inclusion criteria considered in this study were: (1) being affiliated with a sports association, club, or another organisation that promotes PA, and (2) practising any PA regularly. Participants with any medical limitations regarding sub-maximum exercise or that could not understand and follow the assessment protocol of the study were not included [25]. In addition, individuals without medical insurance from the sports association/organisation/club to practice PA did not participate in this study due to legal reasons.

The study was approved by the Scientific Commission of the Department of Physical Education and Sports of the University of Madeira (reference: ACTA n.º 84; 17 January 2017) and the Regional Secretary of Education and Culture. All participants provided informed consent before the assessments, and this study included adherence to the Declaration of Helsinki.

### 2.2. Measures

#### 2.2.1. Educational Level

As a variable related to socioeconomic status, we use educational attainment. Education is a commonly used indicator that measures the life course socioeconomic position. Collected by questionnaire, participants were asked to indicate their highest educational level attained, which was coded into eight levels: (1) no education; (2) 1st cycle; (3) 2nd cycle; (4) 3rd cycle; (5) secondary school level; (6) bachelor’s degree; (7) master’s degree; or (8) PhD degree.

#### 2.2.2. Health-Related Quality of Life

The HRQoL was assessed using the 12-item Short-Form Health Survey (SF-12) [26], which overall includes eight dimensions: physical functioning (PF), role physical (RP), bodily pain (BP), general health (GH), vitality (VT), social functioning (SF), role emotional (RE), and mental health (MH). Two major components were obtained from the sum of scores: (1) physical component (PF + RP + BP + GH), and (2) mental component (VT + SF + RE + MH). Both components’ scores ranged from 0 to 100 each. Higher scores represent a better quality of life related to health. The Portuguese version, including all calculations procedures, can be found in Ribeiro [27].

#### 2.2.3. Physical Activity

The level of PA was assessed using the Baecke questionnaire [28], considering as reference period the last year. This questionnaire included a total of 16 questions classified in 3 dimensions related to PA: (1) PA at work/housework (PA at work), (2) sport (sports PA), and (3) PA during leisure time (this excludes sports during leisure) (leisure-time PA). The total PA index (total PA) was obtained from the sum of these three indices. A detailed description of the scoring procedures for calculation [28] and validation of the Portuguese Baecke questionnaire was previously published [29].

#### 2.2.4. Demographics and Health Profile

Demographic information was collected by questionnaire. Smoking was collected by asking participants whether they smoke or not (yes/no). Alcohol consumption was assessed by the number of drinks (cups) per day usually consumed (1) 0; (2) 1 or 2; (3) 3 or 4; (4) 5 or 6; (5) 7 to 9; (6) >9.

### 2.3. Statistical Analysis

We estimated continuous variables’ mean (M) and standard deviation (SD). Next, we ran a Pearson correlation matrix to observe the raw correlations between the variables under study. Subsequently, we performed mediation analysis specifying two models: (1) to analyse the relationship of education (independent variable) and total PA (mediational variable) on HRQoL (dependent variable); (2) to analyse the relationship between education and PA at work, sports PA, and leisure-time PA and HRQoL (dependent variable). Three dependent variables related to HRQoL were retained for the analysis: the physical component score (PCS) of the SF-12, the mental component score (MCS) of the SF-12, and the total score in the SF-12 questionnaire (SF-12 score). Therefore, both models were run three times, i.e., modifying the dependent variable. All analyses were adjusted for sex, age, smoking, and alcohol use due to their relationship with education, PA, or HRQoL. In the sensitivity analyses of both models, we consider the recoding of the educational level into three levels: (1) <3rd cycle of education, (2) secondary education, (3) higher education. Finally, the single and sequential effect analysis of the mediating variables PA at work and total leisure-time PA were considered (sports PA + leisure-time PA) (model 3). The details of the sensitivity analyses can be found in the online resource. All analyses were also conducted separately in two age groups: (1) 18–44 years old and (2) ≥45 years old.

All analyses were conducted with IBM SPSS v.25. Specifically, to establish the mediation analysis, we used Hayes’s PROCESS macro for SPSS. The macro process proposed by Preacher and Hayes [30] is based on a bootstrapping method that, from the procedure of Baron and Kenny [31], allows for multiple mediations instead of several simple mediations. The bootstrapping method is considered an accurate and powerful mediation method, as it is complemented by the Sobel test to check the validity of the conclusions (it has higher power and better control over type I error), and due to its ability to test the significance of indirect (mediated) effects [32]. To consider the existence of mediation, the following steps were followed: (1) the independent variable was correlated with the mediating variables, establishing the mediating variables as outcome variables; and (2) the mediating variables and the outcome variables were correlated, considering the causal variable as a control variable [33]. To test the statistical significance of indirect effects, bias-corrected 95% confidence intervals (95% CI) were calculated using bootstrapping with 1000 bootstrap samples [34]. When a significant indirect effect was observed, i.e., whose 95% CI did not contain zero, for each considerable mediator, the mediated proportion of the total effect was calculated by dividing the indirect effect by the total effect. In models with more than one mediating variable, we also calculated pairwise contrasts of indirect effects that were significant.

## 3. Results

Regarding the characteristics of the study population (Table 1), 61.2% were women, the average age was 45 years (±15.0 years), and about 63% had at least secondary education. Notably, about 91% were non-smokers, and about 78% drank a maximum of two drinks per day. The highest mean PA score was observed in sports PA (mean = 3.05), with the mean total PA score being 8.56. In the SF-12 scale, for both the MCS and PCS, we observed average scores above 70, with the average total score of the questionnaire being 146.

The correlation matrix (Table 2) indicated that women had a higher age (*p* < 0.001). Education was inversely related to age (*p* < 0.001) and PA at work (*p* < 0.001) and positively related to sports PA (*p* < 0.001). In addition, it was positively related to HRQoL components (PCS: *p* < 0.001, MCS: *p* = 0.016, SF-12 score: *p* < 0.001). Smoking was not related to education and HRQoL. However, it was inversely related to sports PA and total PA (*p* < 0.001). Alcohol use was not related to PA-related variables, but it was positively associated with education (*p* = 0.019) and inversely related to HRQoL (PCS: *p* < 0.001, MCS: *p* = 0.003, and SF-12 score: *p* < 0.001). All PA-related variables (except the relationship between PA at work and sports PA (*p* > 0.05)) indicated a positive and statistically significant relationship. In addition, sports PA and total PA were positively related to all three variables related to HRQoL (in all three components: *p* < 0.001 for sports PA; PCS and SF-12 score: *p* < 0.001; and MCS: *p* = 0.002 for total PA), as well as leisure-time PA for MCS (*p* = 0.011) and SF-12 score (*p* = 0.022). The three HRQoL components were also positively related (*p* < 0.001).

In Table 3, we found significant indirect effects of educational level on HRQoL (considering PCS, MCS, and SF-12 score) through total PA (model 1). Specifically, we observed inconsistent mediation, i.e., total PA acted as a suppressor variable. Thus, the total effect of education on HRQoL (considering PCS, MCS, and SF-12 score) was smaller than the direct effect, as both effects tended to cancel out. In the relationship between education and MCS, the total and direct effects were not different from zero (Figure 1). However, the suppression effect (indirect effect: b = −0.1275; 95% CI −0.2358, −0.0324) was observed (Table 3). Indeed, the correlation between education and MCS adjusted for covariates had low power. In contrast, the correlations between education and total PA, and total PA and MCS, were non-trivial. The proportion suppressed was 9.5%, 38.5%, and 15.7%, considering PCS, MCS, and SF-12 score as dependent variables, respectively. In model 1 coding education into three categories (Appendix A), the proportion suppressed was 8.4%, 45.9%, and 14.8% (Appendix A).

Model 2 showed that the direct effect was significant for PCS and SF-12 score, but not for MCS (Figure 2), and that the difference between the total and direct effect, the total indirect effect, was not (total indirect effect: b = 0.0574 (95% CI −0.0280, 0.1671) for PCS, b = 0.0013 (95% CI −0.1421, 0.1425) for MCS, and b = 0.0587 (95% CI −0.1750, 0.2863) for SF-12 score) (Table 3). The individual indirect effects, including only PA at work and leisure-time PA as mediating variables, were not a non-zero indirect effect. They were not related to PCS, MCS, and SF-12 score. However, it is worth noting some differences that emerged considering PCS and SF-12 score as outcome variables in the individual indirect effects. Thus, the suppression effect was also observed in model 2; specifically on the pathway linking education and PCS and SF-12 score mediated by PA at work and sports PA (indirect effect: b = −0.0219 (95% CI −0.0487, −0.0030) for PCS and b = −0.0356 (95% CI −0.0810, −0.0034) for SF-12 score) (Table 3). However, the pathway including only sports PA mediated the association between education and SF-12 score and PCS (indirect effect: b = 0.0799 (95% CI 0.0211, 0.1606) for PCS and b = 0.1296 (95% CI 0.0268, 0.2811) for SF-12 score) (Table 3). The proportion mediated (considering only sports PA as a mediating variable) was 12.5% and 15.4% when PCS and SF-12 score were dependent variables. In model 2 coding education into three categories (Appendix A), the mediated proportion was also observed for MCS (15.6%, 50.2%, and 21.4% for PCS, MCS, and SF-12 score, respectively) (Appendix A). In contrast, the suppressed proportion for sequencing PA at work and sports PA mediators was 3.4% for PCS and 4.4% for SF-12 score—coding education into three categories: 3.9% and 5.3%, respectively. Furthermore, when examining comparisons of indirect effects that did not contain zero and, therefore, were significant, mediation through sports PA as the sole mediating variable was significantly stronger than the suppression observed in the pathway containing the variables PA at work and sports PA (indirect effect: b = 0.1018 (95% CI 0.0317, 0.1990) for PCS and b = 0.1652 (95% CI 0.0384, 0.3190) for SF-12 score). We observed similar findings by coding education into three categories (indirect effect: b = 0.2241 (95% CI 0.0861, 0.3793) for PCS and b = 0.3707 (95% CI 0.1202, 0.6926) for SF-12 score.

In the sensitivity analysis corresponding to model 3 (Appendix A) coding education into three categories, we observed an indirect suppression effect when considering the path sequences of the mediating variables PA at work and total-leisure-time PA. The proportion suppressed was 3.4%, 13.4%, and 5.5%, for PCS, MCS, and SF-12 score, respectively, and coding education into three categories, 3.6%, 19.1%, and 6.4% (Appendix A).

In the analyses by age, in the population aged between 18 and 44 years old, in model 1 we also observed a suppression effect of total PA on the relationship between SES and the three dependent variables related to HRQoL (i.e., PCS, MCS, and SF-12 score) (Appendix A) and between SES and PCS and SES and SF-12 score by classifying education into three categories (Appendix A). Concerning model 2, coding education into three categories, for PCS we observed a significant total indirect effect (total indirect effect: b = 0.3153 (95% CI 0.0175, 0.6283)) (Appendix A). Individual indirect effects that included only sport PA also had a non-zero indirect effect for PCS, MCS, and SF-12 score as outcome variables (b = 0.2892 (95% CI 0.0730, 0.5335) for PCS; b = 0.2759 (95% CI 0.0384, 0.5901) for MCS; b = 0.5651 (95% CI 0.1611, 1.0813) for SF-12 score). In contrast, the suppression effect was also observed: in the pathway containing sports PA and leisure-time PA for PCS (b = −0.0489 (95% CI −0.1323, −0.0005)) and in the pathway containing PA at work and sports PA for SF-12 score (b = −0.1666 (95% CI −0.4376, −0.0009)). However, when examining comparisons of indirect effects that did not contain zero and, therefore, were significant, mediation through sports PA as the sole mediating variable was significantly stronger than the suppression observed in the pathway containing the variables sports PA and leisure-time PA for PCS (indirect effect: b = 0.3381 (95% CI 0.0884, 0.6748)) and the pathway containing the variables PA at work and sports PA for SF-12 score (indirect effect: b = 0.7316 (95% CI 0.2088, 1.4301)). In model 3, we observed no significant indirect effects in this age group (Appendix A).

In the population ≥ 45 years old, we only observed a significant indirect effect in model 1 considering PCS as the dependent variable: total PA suppressed the relationship between educational level and PCS by 10.4% (Appendix A). However, significant indirect effects were not observed when coding education into three categories (Appendix A). Concerning model 2, we found no significant indirect effects (Appendix A), but we also observed an indirect suppression effect when considering the path sequences of the mediating variables PA at work and total leisure-time PA (model 3). The proportion suppressed was 4.1% and 7.6%, for PCS and SF-12 scores, respectively, and coding education into three categories, 4.5% and 9.5%, respectively (Appendix A). More details on the analyses separated by age groups can be found in the online resources (Appendix A).

## 4. Discussion

We found that total PA and sports PA mediated the relationship between educational level and HRQoL. Specifically, we observed an inconsistent mediation, i.e., a suppression effect (see [35]) of total PA on the impact of educational level on HRQoL. Therefore, total PA declined the magnitude of the relationship between SES and HRQoL. In contrast, sports PA was a mediator variable, which implied that educational level was related to HRQoL through the sports PA. Total PA suppressed the socioeconomic gradient of HRQoL by 8–10%, 39–46%, and 15–16%, respectively, for the PCS, MCS, and SF-12 score, while sports PA mediated the socioeconomic gradient of HRQoL by 13–16%, 50%, and 15–21% for PCS, MCS, and SF-12 score.

Educational level was positively related to HRQoL, particularly with the PCS and SF-12 score. However, using bivariate analyses and different SES indicators, previous studies have also found a consistent association between SES and different measures of HRQoL [5,36,37,38,39]. In our study, we also provide evidence of the total, direct, and indirect effects of SES on HRQoL by considering different domains of PA. In this context, it was observed that total PA and sports PA are factors that determine the relationship between SES and HRQoL. However, this effect was partial and did not fully explain the relationship between SES and HRQoL.

As single mediating variables, PA at work and leisure-time PA were unrelated to HRQoL, despite their inverse relationship with SES. In addition, both PA at work and leisure PA did not influence the relationship between SES and HRQoL. To the best of our knowledge, this is the first study that has analysed the effect of domain-specific PA on the relationship between SES and HRQoL. In past literature, Rezaei et al. [39], through a Blinder–Oaxaca Decomposition, demonstrated that low PA (14.4%), having a chronic disease (13.9%), and smoking (11.4%) contributed to the overall difference in the prevalence of poor HRQoL—assessed using EuroQol—between the poorest and wealthiest SES groups in Iranian adults. Our results also complement previous studies that have determined the relationship between SES and well-being using path analysis. For example, Bielderman et al. [40] examined the relationship between QoL—using the CASP-19 questionnaire—and SES, physical function, social functioning, depressive symptoms, and self-efficacy in older adults. The authors observed an indirect effect of SES on QoL by social functioning, depressive symptoms, and self-efficacy. Furthermore, Ho-Joong et al. [41] examined the effect of frailty on the relationship between SES and HRQoL—using the EQ-5D—and concluded that both income and education were related to HRQoL through frailty.

The literature suggests that leisure-time PA is more common among people with higher SES, mainly due to self-efficacy and social support, which explain almost all differences related to PA, as well as physical barriers and lack of accessibility to public spaces, which may also partially explain differences related to transport-related PA [42]. However, although leisure-time PA and sports PA were directly related in our results, sports PA was more common with increasing educational attainment. In contrast, leisure-time PA was more likely in those with lower educational attainment. Therefore, our results support that people with higher SES engage in higher-intensity activities or exercises [43], which is where the highest mean scores are observed in this sample. At the same time, leisure-time PA, which is related to other specific domains (i.e., PA related to transport or unstructured cycling or walking outings), could be higher among people with lower SES [43]. Moreover, the fact that people with a higher educational level perform more sports PA could be related, in turn, to the greater PA at work or total PA performed by people with a lower SES, which could condition preferences for spending leisure time or the characteristic for structured exercise.

There could be several reasons why people with lower SES have higher total PA in this context. Indeed, people with lower educational attainment usually work in jobs with a high level of manual labour [44], which could explain their higher PA at work. In addition, the greater sedentary behaviour of people with higher education could contribute to these results. There is evidence that a higher SES is positively associated with sitting time [44,45,46,47,48], mainly due to the job characteristics [44,45,48], even though the literature has described that people with a higher SES are more engaged in leisure-time PA, comply more frequently with the WHO moderate-to-vigorous PA guidelines [23,44,49,50], and tend to indicate fewer barriers to PA [50]. Thus, it is likely that PA preferences, conditioned by the interaction or environment in which the activity takes place and the greater number of options, time, and resources available to the high-SES sub-population for PA, may influence these outcomes [43,51].

Additionally, our results support previous research claiming that total PA and sports PA are related to HRQoL [17,18,19], although we did not observe this relationship for leisure PA. It is worth noting that the highest leisure-time PA in the population analysed comes from sports PA. In this context, it should be borne in mind that the population analysed are adults who regularly practice PA in sports or leisure centres or programmed exercise. Leisure PA is not related to HRQoL, while sports PA is related to all three components of HRQoL, which could be associated with the practice of higher-intensity PA, such as structured sports and activities that make them sweat. In this context, the results could argue that people with a better HRQoL can engage in more vigorous PA. Furthermore, unlike other studies showing an inverse relationship between PA at work and measures of HRQoL [20], in this study, we did not observe a statistically significant relationship between the two variables. However, PA at work seems to be related to the HRQoL only if we consider leisure-time PA. Proof of this is that PA at work sequenced by sports PA or total leisure-time PA was positively related to HRQoL.

Several limitations must be considered in the discussion of this work. Firstly, this is a cross-sectional observational design. Therefore, it precluded causal relationships among variables. All variables analysed were assessed at a single point in time, and we cannot establish a cause–effect relationship in the relationship between variables. Secondly, all variables were evaluated through questionnaires based on self-reporting. Possible recall biases from the data collection procedure must be considered. Considering that the surveyed population belongs to sports centres and clubs, a limitation of this work could be related to an under-representation of the people that usually do not perform physical activity in their leisure time. Therefore, considering this data collection procedure, the results should be interpreted in the context population affiliated with the Madeira Association of Sport for All in the Autonomous Region of Madeira. Despite these limitations, it should be noted that to the best of our knowledge, this is the first study that has demonstrated the mediating effect of different domains of PA on the relationship between educational attainment and HRQoL. Therefore, a strength of this work is that our PA assessment includes household chores and activity at work and distinguishes leisure-time PA that is more related to transport or rides (biking or walking) from higher-intensity sports activities.

## 5. Conclusions

Our results conclude that sports PA mediates the relationship between SES and HRQoL; in contrast, work PA and leisure-time PA, as single mediators, are not mediating variables between SES and HRQoL. The results add that total PA suppresses the socioeconomic gradient in HRQoL. PA should be taken into account when designing effective health promotion programmes, taking into account that specific domains of PA demonstrate different socioeconomic gradients of HRQoL. Thus, assuming that people with lower educational attainment have lower HRQoL scores, health promotion policies should consider possible individual, economic, environmental, and social barriers to exercise and programmed sports among the most vulnerable people. Future works should consider the development of longitudinal studies to assess whether these results are sustained over time. Furthermore, given that we have only observed partial suppression and mediation effects, these longitudinal studies should also include other health-related lifestyle factors that may influence the relationship between SES and HRQoL.

## Figures and Tables

**Figure 1 ijerph-19-07608-f001:**
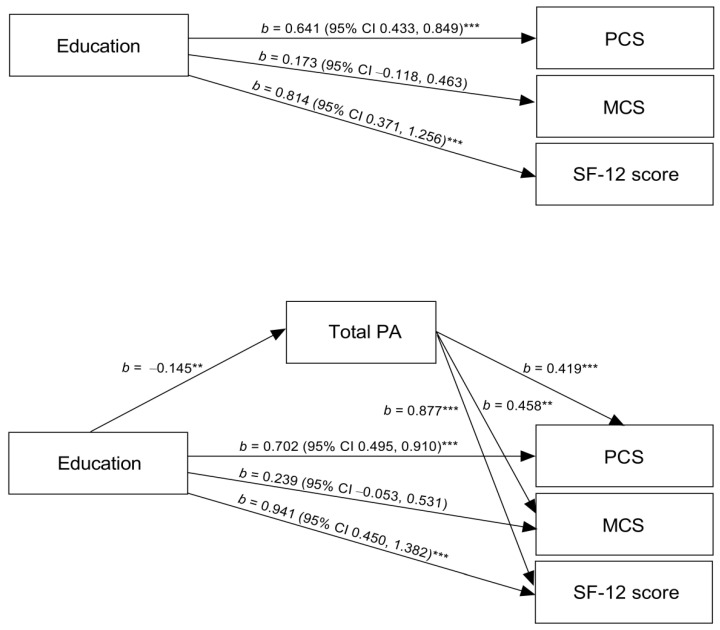
Relationship between educational attainment and total physical activity in health-related quality of life (N = 364). We show unstandardized regression coefficients (b) and bootstrap confidence intervals for the association. PA is physical activity. PCS and MCS are physical and mental component summaries of SF-12. SF-12 score corresponds to the total score on the SF-12 scale. ** *p* ≤ 0.01, *** *p* < 0.001.

**Figure 2 ijerph-19-07608-f002:**
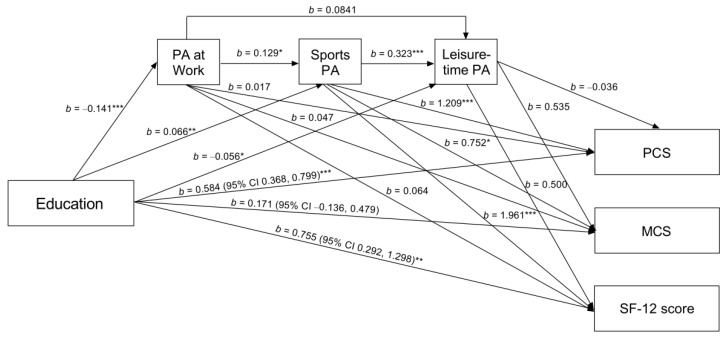
Relationships between educational attainment and work, sport, and leisure-time physical activity in health-related quality of life (N = 364). We show unstandardized regression coefficients (b) and bootstrap confidence intervals for the association. PA is physical activity. PCS and MCS are physical and mental component summaries of SF-12. SF-12 score corresponds to the total score on the SF-12 scale. * *p* < 0.05, ** *p* ≤ 0.01, *** *p* < 0.001.

**Table 1 ijerph-19-07608-t001:** Descriptive analysis of the studied variables in the adult population of Madeira.

Variables	N Total	N (%)	Mean (SD)
**Sex**			
Men	381	148 (38.8)	
Women		233 (61.2)	
**Age**	381		45 (15)
**Educational attainment**			
No schooling	380	10 (2.6)	
First cycle		57 (15.0)	
Second cycle		25 (6.6)	
Third cycle		48 (12.6)	
Secondary Education		117 (30.8)	
Bachelor’s degree		96 (25.3)	
Master’s degree		23 (6.1)	
PhD degree		4 (1.1)	
**Current smoker**			
No	381	349 (91.6)	
Yes		32 (8.4)	
**Daily alcohol drink**			
0 cups/daily	376	93 (24.7)	
1 or 2 cups/daily		199 (52.9)	
3 or 4 cups/daily		55 (14.6)	
5 or 6 cups/daily		14 (3.7)	
7 or 9 cups/daily		6 (1.6)	
>9 cups/daily		9 (2.4)	
**PA at work (1–5 units)**	381		2.85 (0.67)
**Sports PA (1–5 units)**	381		3.05 (0.67)
**Leisure-time PA (1–5 units)**	381		2.66 (0.62)
**Total PA (3–15 units)**	381		8.56 (1.34)
**PCS (0–100 pts)**	370		71.57 (22.10)
**MCS (0–100 pts)**	370		74.80 (19.34)
**SF-12 score (200 pts)**	370		146.37 (36.72)

PA is physical activity. PCS and MCS are physical and mental component summaries of SF-12. SF-12 score corresponds to the total score on the SF-12 scale.

**Table 2 ijerph-19-07608-t002:** Correlation matrix of the studied variables in the adult population of Madeira.

Variables ^†^	1	2	3	4	5	6	7	8	9	10	11
1 Women	1.00										
2 Age	0.324 ***	1.00									
3 Education	−0.149 **	−0.469 ***	1.00								
4 Smoking	−0.186 ***	−0.106 *	0.030	1.00							
5 Alcohol use	−0.300 **	−0.208 ***	0.121 *	0.225 ***	1.00						
6 PA at work	−0.013	0.006	−0.234 ***	0.001	0.022	1.00					
7 Sports PA	−0.198 ***	−0.207 ***	0.190 ***	−0.184 ***	0.006	0.080	1.00				
8 Leisure-time PA	−0.154 **	−0.082	−0.058	−0.086	0.051	0.163 **	0.354 ***	1.00			
9 Total PA	−0.177 ***	−0.139 **	−0.049	−0.132 **	0.038	0.618 ***	0.707 ***	0.722 ***	1.00		
10 PCS	−0.156 **	−0.423 ***	0.444 ***	0.061	0.170 **	−0.033	0.312 ***	0.074	0.175 ***	1.00	
11 MCS	−0.150 **	−0.152 **	0.126 *	0.029	0.103 *	0.017	0.184 ***	0.132 *	0.162 **	0.568 ***	1.00
12 SF-12 score	−0.172 ***	−0.309 ***	0.303 ***	0.049	0.150 **	−0.006	0.272 ***	0.119 *	0.189 ***	0.860 ***	0.909 ***

PA is physical activity. PCS and MCS are physical and mental component summaries of SF-12. SF-12 score corresponds to the total score on the SF-12 scale. * *p* < 0.05; ***p* ≤ 0.01; *** *p* < 0.001. ^†^ The numbers in the columns match the numbers in the rows, identifying each variable.

**Table 3 ijerph-19-07608-t003:** Relationships between educational attainment and PA in health-related quality of life (N = 364).

Pathways Model 1 Key	Indirect Effect	PM or PS
EDUC → Total PA → (outcome PCS)	−0.0609 (95% CI −0.1178, −0.0186)	PS = −9.5%
EDUC → Total PA → (outcome MCS)	−0.0666 (95% CI −0.1418, −0.0113)	PS = −38.5%
EDUC → Total PA → (outcome SF-12 score)	−0.1275 (95% CI −0.2358, −0.0324)	PS = −15.7%
**Pathways Model 2 key**		
**Outcome PCS**		
Total (model)	0.0574 (95% CI −0.0280, 0.1671)	
EDUC → PA at work → PCS	−0.0024 (95% CI −0.0651, 0.0667)	
EDUC → Sports PA → PCS	0.0799 (95% CI 0.0211, 0.1606)	PM = 12.5%
EDUC → Leisure-time PA → PCS	0.0020 (95% CI−0.0292, 0.0379)	
EDUC → PA at work → Sports PA → PCS	−0.0219 (95% CI −0.0487, −0.0030)	PS = −3.4%
EDUC → PA at work → Leisure-time PA → PCS	0.0004 (95% CI −0.0075, 0.0075)	
EDUC → Sports PA → Leisure-time PA → PCS	−0.0008 (95% CI −0.0138, 0.0114)	
EDUC → PA at work → Sports PA → Leisure-time PA → PCS	0.0002 (95% CI −0.0029, 0.0040)	
**Outcome MCS**		
Total (model)	0.0013 (95% CI −0.1421, 0.1425)	
EDUC → PA at work → MCS	−0.0067 (95% CI −0.1179, 0.0967)	
EDUC → Sports PA → MCS	0.0497 (95% CI −0.0001, 0.1244)	
EDUC → Leisure-time PA → MCS	−0.0301 (95% CI −0.0837, 0.0090)	
EDUC → PA at work → Sports PA → MCS	−0.0137 (95% CI −0.0368, 0.0008)	
EDUC → PA at work → Leisure-time PA → MCS	−0.0063 (95% CI −0.0213, 0.0028)	
EDUC → Sports PA → Leisure-time PA → MCS	0.0114 (95% CI −0.0037, 0.0343)	
EDUC → PA at work → Sports PA → Leisure-time PA → MCS	−0.0031 (95% CI −0.0096, 0.0014)	
**Outcome SF-12 score**		
Total (model)	0.0587 (95% CI −0.1750, 0.2863)	
EDUC → PA at work → SF-12 score	−0.0091 (95% CI −0.1731, 0.1578)	
EDUC → Sports PA → SF-12 score	0.1296 (95% CI 0.0268, 0.2811)	PM = 15.4%
EDUC → Leisure-time PA → SF-12 score	−0.0281 (95% CI −0.1097, 0.0305)	
EDUC → PA at work → Sports PA → SF-12 score	−0.0356 (95% CI −0.0810, −0.0034)	PS = −4.4%
EDUC → PA at work → Leisure-time PA → SF-12 score	−0.0059 (95% CI −0.0310, 0.0073)	
EDUC → Sports PA → Leisure-time PA → SF-12 score	0.0107 (95% CI −0.0111, 0.0438)	
EDUC → PA at work → Sports PA → Leisure-time PA → SF-12 score	−0.0029 (95% CI −0.0122, 0.0035)	

PA is physical activity. PCS and MCS are physical and mental component summaries of SF-12. SF-12 score corresponds to the total score on the SF-12 scale. Proportion mediated (PM) or proportion suppressed (PS) represent the percentage of the total effect of educational attainment on health-related quality of life by each pathway.

## Data Availability

Data are available under request.

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
