# Peer review of "Exploring the Role of Physical Activity in Mediating the Association between Educational Level and Health-Related Quality of Life in an Adult Lifespan Sample from Madeira Island"

_ijerph, 2022, doi:10.3390/ijerph19137608_

Round 1

Reviewer 1 Report

The relationship between physical activity and quality of life has long been established, so I was curious to see how the authors would provide new insight to this area of research. I was very impressed with the statistical modeling and detailed descriptions of how the various physical activity domains influence the effect of SES on HRQoL. It was also interesting to see that SES was predictive of they types of physical activity that individuals performed. I was a bit concerned about the wide age distribution and think you should have compared age groups separately. I would recommend doing this to see if there were any differences and reporting this in the final draft. 

Author Response

Thank you very much for your flattering comment. Thank you also for your suggestion, which has contributed to making this work scientifically sounder. All analyses were also conducted separately in two homogeneous age groups in terms of population size: (1) 18-44 years, (2) 45 years and older. These analyses can be found in the supplementary resources (Tables S3 to S9). We hope that these new analyses are in line with your request.

Reviewer 2 Report

Overall, the background and discussion appear sound. I also appreciate the discussion of the limitations of the study itself. Conducting a study where you evaluate physical activity by recruiting at an athletic club makes me think you are likely missing a large part of the population, limiting the generalizability of findings. I also feel that there are leaps made between the analysis and the conclusions that I think should be spelled out more clearly. Taken altogether, the paper adds a small piece of information to the literature but its actual significance needs more addressing than what the current presentation provides.

Author Response

Thank you for your comment. 

In response to your first suggestion, we have included in the limitations of the paper that the sample comes from Madeiran residents participating in sports clubs or centres and therefore the analyses should not be generalised to the Madeiran population. 

Secondly, to avoid jumps between analyses and conclusions, we have reduced the conclusions section to limit ourselves to talking about the mediation aspect between SES and HRQoL through physical activity, which is the focus of the study. In addition, we have added new analyses to the paper, analysing the results by age group, which we believe may provide more robustness to the findings. These analyses can be found in the supplementary material. We hope these changes will be to your satisfaction.

Reviewer 3 Report

Dear Authors,

I have some suggestions:

The meaning of all abbreviations must be included in the main document and in the abstract, examples WHO, SF-12….. beside once they are indicated used them, examples: socioeconomic status, physical activity..  are writing in the document after abbreviation explanation. The PA is explained 2 times, delete the second one.

All tables and figures must have the meaning of all abbreviation

The table of correlation is table 2, no 1.

In the abstract the results and conclusion should be more clear and specific, related with the title

In the main document, the conclusion is also too long, please to synthesize and present the most relevant information.

The SES varied a lot between people aged 18 to 89, why such a wide range? It could be another limitation to include all of them in the same group

“while sports PA mediated the socioeconomic gradient of HRQoL by 13-16%, 50%, 265 and 15-21% for PCS, MCS, and SF-12 score.” I think a respectively is missing at the end

Best Regards

Author Response

Dear reviewer, we appreciate your suggestions. We have responded to each of them and the corresponding changes have been implemented in the new version of the article. 
1. All abbreviations have been included in the abstract and we have removed the explanation of PA that appeared for the second time in the methodological section.
2. Following your suggestion, we have added the meanings of the abbreviations in all tables and figures.
3. The correlation table is now table 2.
4. Taking into account the title and aim of the article, the results of the article summary and conclusions now only include information on the mediation of specific domains of physical activity in the relationship between SES and HRQoL. We hope you will be pleased with these changes.
5. In response to your comment, the conclusion has been shortened and we have focused on the mediation of physical activity in the relationship between SES and HRQoL. The first four lines are the conclusions of the paper, the rest are recommendations based on these findings that we believe may be relevant to the reader. We hope that these changes are in line with your request.
6. Thank you for this comment. Following your suggestion on age groups, all analyses have been conducted separately in two age groups. These analyses have been included in the supplementary material as additional sensitivity analyses. We hope you are satisfied with these changes.
7. The word "respectively" has now been included.